# HIV virologic response, patterns of drug resistance mutations and correlates among adolescents and young adults: A cross-sectional study in Tanzania

Joan Rugemalila[1,2]*, Doreen Kamori[1], Peter Kunambi[1], Mucho Mizinduko[1], Amon Sabasaba[1], Salim Masoud[1], Frank Msafiri[1], Sabina Mugusi[1], Rita Mutagonda[1], Linda Mlunde[1], Davis Amani[1], Erick Mboya[1], Macdonald Mahiti[1], George Ruhago[1], Jeremiah Mushi[3], Veryeh Sambu[3], George Mgomella[4], Boniface Jullu[5†], Werner Maokola[3], Prosper Njau[3], Beatrice Mutayoba[3], Godfrey Barabona[6], Takamasa Ueno[6], Andrea Pembe[1], Tumaini Nagu[1], Bruno Sunguya[1,6], Said Aboud[1]

**1** Department of Microbiology and Immunology, Muhimbili University of Health and Allied Sciences, Dar es Salaam, Tanzania, **2** Department of Internal Medicine, Muhimbili National Hospital, Dar es Salaam, Tanzania, **3** National AIDS Control Programme, Dodoma, Tanzania, **4** Centers for Disease Control and Prevention, Dar es Salaam, Tanzania, **5** Management and Development for Health, Dar es Salaam, Tanzania, **6** Joint Research Center for Human Retrovirus Infection, Kumamoto University, Kumamoto, Japan

† Deceased.

* joanrugemalila@gmail.com

**Data Availability Statement:** All relevant data are within the paper and its Supporting Information file (data set).

## Abstract

### Background

The emergence of HIV drug resistance mutations (DRMs) is of significant threat to achieving viral suppression (VS) in the quest to achieve global elimination targets. We hereby report virologic outcomes and patterns of acquired DRMs and its associated factors among adolescents and young adults (AYA) from a broader HIV drug resistance surveillance conducted in Tanzania.

### Methods

Data of AYA was extracted from a cross-sectional study conducted in 36 selected facilities using a two-stage cluster sampling design. Dried blood spot (DBS) samples were collected and samples with a viral load (VL) ≥1000 copies/mL underwent genotyping for the HIV-1 *pol* gene. Stanford HIV database algorithm predicted acquired DRMs, Fisher's exact test and multivariable logistic regression assessed factors associated with DRMs and VS, respectively.

### Findings

We analyzed data of 578 AYA on antiretroviral therapy (ART) for 9–15 and ≥ 36 months; among them, 91.5% and 88.2% had VS (VL<1000copies/mL) at early and late time points, respectively. Genotyping of 64 participants (11.2%) who had VL ≥1000 copies/ml detected 71.9% of any DRM. Clinically relevant DRMs were K103N, M184V, M41L, T215Y/F,

**Funding:** The HIV Global Fund for Malaria, Tuberculosis and HIV/AIDS in Tanzania provided funding for the national ADR survey through the Ministry of Health Community Development Gender Elderly and Children". Additionally, JR performed this research work as part of post graduate studies and, received a scholarship from the Swedish International Development Agency (SIDA) for research courses. The funders had no role in study design, data collection and analysis, decision to publish, or preparation of the manuscript.

**Competing interests:** The authors have declared that no competing interests exist.

L210W/L, K70R, D67N, L89V/T, G118R, E138K, T66A, T97A and unexpectedly absent K65R. Participants on a protease inhibitor (PI) based regimen were twice as likely to not achieve VS compared to those on integrase strand transfer inhibitors (INSTI). The initial VL done 6 months after ART initiation of $\geq$1000copies/mL was the primary factor associated with detecting DRMs ($p$ = .019).

## Conclusions

VS amongst AYA is lower than the third UNAIDs target. Additionally, a high prevalence of ADR and high levels of circulating clinically relevant DRMs may compromise the long-term VS in AYA. Furthermore, the first VL result of $\geq$1000copies/ml after ART initiation is a significant risk factor for developing DRMs. Thus, strict VL monitoring for early identification of treatment failure and genotypic testing during any ART switch is recommended to improve treatment outcomes for AYA.

## Introduction

The expanded access to antiretroviral therapy (ART) has proved to be an effective intervention in improving quality of life and reducing mortality among people living with HIV (PLHIV) [1]. However, despite comprehensive ART coverage, the burden of new HIV infections is the highest in Sub-Saharan Africa [2], where the majority of adolescents living with HIV reside [3]. Moreover, young women and adolescent girls account for 25% of all new HIV infections globally [4, 5]. This unprecedented burden is higher compared to adolescent boys [6, 7]. The efforts to ensure early initiation to effective ART has been implemented with the test and treat strategy, but treatment adherence has remained a challenge among AYA [8, 9]. As a result, a low level of VS in AYA further predisposes them to a significant risk of developing acquired drug resistance (ADR) [10–13].VS among adolescents is lower than in adults, ranging from below 50% to 80% [3, 9, 10, 14, 15]. HIV drug resistance (HIVDR) is a substantial barrier to reaching the UNAIDS Fast-Track goal of ending AIDS by 2030 [16, 17] because is associated with poor clinical outcomes and reduces ARV effectiveness compromising the third UNAIDS target.

The burden of ADR has been growing gradually over time on commonly used ART. A significant threat in East and Southern Africa is driven by nonnucleoside reverse transcriptase inhibitors (NNRTI) resistance [18–21]. Notably, increasing multi-class acquired drug resistance has been widely reported [10, 11, 22, 23], as well as increasing NNRTI resistance amongst newly-diagnosed infants [24], showing the growing burden of drug resistance. Newer drugs in the class integrase strand transfer inhibitors (INSTI) such as dolutegravir have had very low to no drug resistance [25]. Nevertheless, suboptimal adherence and virologic monitoring with limited access to routine drug resistance mutation testing may increase the chances of their resistance. The burden and patterns of HIV-DRMs of clinical importance vary geographically and in different HIV-infected subpopulations. Importantly, the world health organization (WHO) reports a scarcity of data informing the magnitude of DRMs in children and adolescents in SSA. Since adolescents face unique challenges including but not limited to developmental dynamics and cognitive behavior in the context of HIV services delivery, focused reporting on HIV-DRM among adolescents and young adults is fundamental rather than covering them under general PLHIV.

In the recent years, the increasing prevalence of NNRTI resistance mutations in ART-naive and experienced individuals led to policy recommendations to adopt INSTIs based regimens

[11, 26]. Accordingly, dolutegravir (DTG), in combination with tenofovir (TDF) and lamivudine (3TC), a single fixed dose combination tablet (TLD) has now replaced efavirenz (EFV)-based first-line regimens in Tanzania. PLHIV on TLE transitioned to TLD on 18 March 2019 onwards regardless of VS status, an approach that may have risked development of DRMs selecting for NRTIs backbone from unrecognized non-viral suppression. To generate estimates on VS, patterns of HIV-DRMs, and its associated factors among AYA in Tanzania, we successfully used viral load and genotyping data from a broader national surveillance of ADR conducted among children and adults initiated on ART, one year after the rollout of Dolutegravir.

## Materials and methods

### Study design and settings

This study used data from the national ADR survey (Fig 1) conducted on the Tanzania mainland between July and October 2020 [27]. We included 36 care and treatment centres (CTCs) selected from 982 facilities representing 90% of PLHIV attending CTCs in Tanzania. Details of selection and sampling are explained in the protocol paper [27].

In 2015, Tanzania adopted the "test and treat" strategy the World Health Organization recommended, making all people living with HIV eligible for treatment after HIV diagnosis.

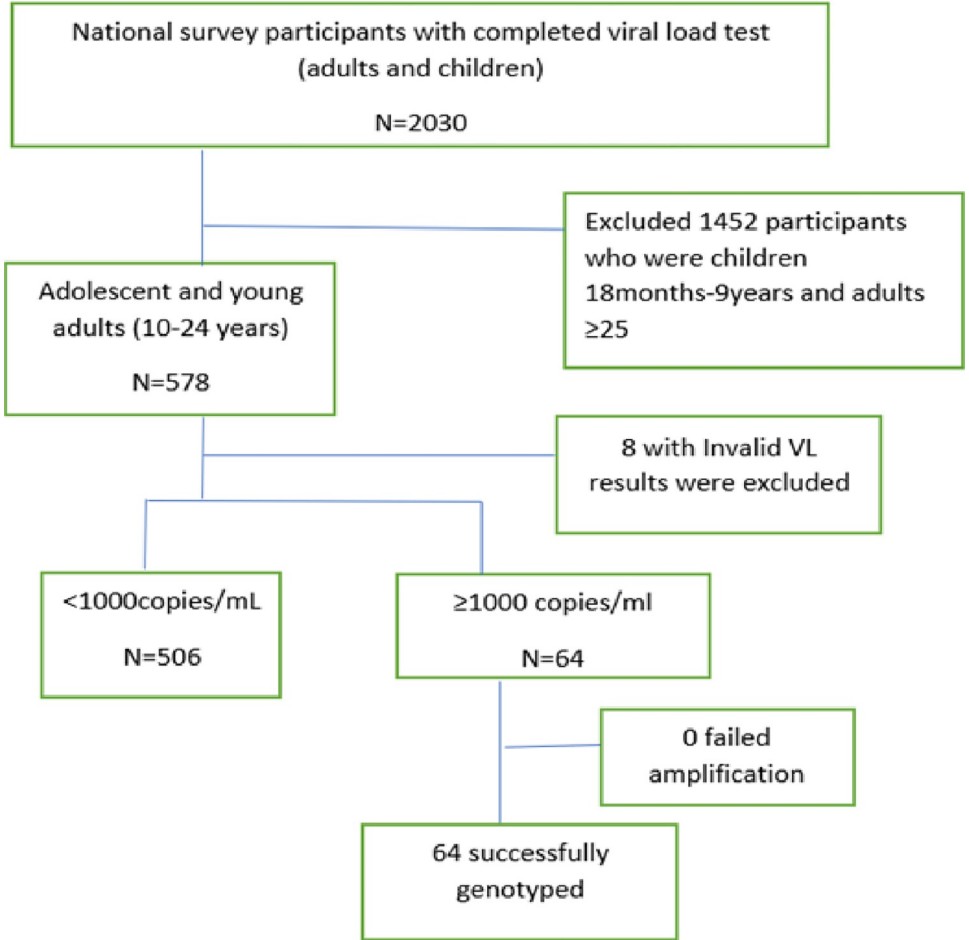

**Fig 1. Flow chart showing enrollment of study participants, viral load and genotypic testing.**

Later, routine VL testing for all clients on ART, regardless of age or disease stage, was adopted in 2017. Furthermore, in 2018, the national package on adolescents living with HIV and AIDS management and the transition to adult care was adopted. As a result, AYA receives HIV care and treatment services according to their age group; young adolescents 10–14 years in pediatrics clinics, older adolescents 15–19 years and youth 20–24 years receive care in adolescent and youth-friendly clinics. Moreover, the adolescent and youth-friendly clinics comprise a separate space and privacy for AYA, special times when AYA can receive services, convenient hours, youth-friendly surroundings and availability of peer educators.

## Study procedures

Eligible participants aged 18–24 years and 15–17 years provided written informed consent and assent, respectively. In addition, we obtained informed consent from parents or legal guardians of children and adolescents aged 10–17 years. Furthermore, we retrieved social demographic information (age, sex, highest education level attained) and clinical characteristics, including a history of ART (treatment initiation date, past and current ART regimens, ART duration) from the medical records from each study site. At the time of survey, early time point assessment for VS included participants receiving ART for 9–15 months and late time point among those on ART for more than 36 months as per the WHO guidance [28]. One VL measurement from plasma samples was performed in real-time using the existing VL monitoring system in Tanzania, and we defined VS as VL<1000 copies of viral RNA/ml of blood. The survey coordinator issued the results to the respective sites for clinical management of participants according to the current national ART treatment guidelines. The procedures for sample collection, processing, quality assurance and sample storage in the microbiology laboratory at Muhimbili University of Health and Allied Sciences have been described elsewhere [27].

## HIV drug resistance mutation genotyping

HIVDR genotyping was primarily done from DBS samples and plasma was used as a backup for samples that were not successfully sequenced from DBS. We subjected DBS samples from participants with completed survey questionnaires and having VL results ≥1000 copies/mL to genotypic testing at the WHO accredited laboratory in British Columbia, Canada. HIV RNA was first extracted from DBS samples using the Nested reverse transcriptase-polymerase chain reaction (RT-PCR) protocol [29]. The next step was the amplification of reverse transcriptase, protease-reverse transcriptase (PR-RT), and integrase (IN) regions of the HIV *pol gene* using Applied Biosystems GeneAmp PCR System 9700 or VeritiPro Thermal Cycler. Sets of routine outer and inner forward primers; `5'PROT1: TAATTTTTAGGGAAGAT CTGGCCTTCC; AGTAGGACCTACACCTGTCA; 5'PROT2: TCAGAGCAGACCAGAGCCAACAGCCCC;` and `A35: TTGGTTGCACTT TAAATTTTCCCATTAGTCCTATT` and reverse primers `3'PROT1: GCAAATACTGGAGTATTGTATGGATTTTCAGG; MJ4: CTGTTAGTGCTT TGGTTCCTCT; 3'PROT2: AATGCTTTTATTTTTTCTTCTGTCAATGGC;` and `3' NE135: CCTACTAAC TTCTGTATGTCATTGACAGTCCAGCT` for PR-RT region were used for PCR amplification and sequencing. Whereas, for IN region the forward primers `INPS1: TAGTAGCCAGCTGT GATAAATGTC` and `INPS3: GAAGCCATGCATGG CAAG;` and reverse primers `INPR8: TTCCATGTTCTAATCCTCATC CTG` and `INPR9: ATCCTCATCCTGTCTACT TGCC` were used. A set of inner MJ3 PCR primers for PR-RT and IN regions were used for Sanger sequencing using Applied Biosystems 3730XL DNA Analyser according to manufacturer's instructions.

Furthermore, sequences were imported for analysis to Seqscape software version 2.7 and the Gene cutter tool; in the Los Alamos sequence database (https://www.hiv.lanl.gov/content/sequence/GENE_CUTTER/cutter.html). We identified mutations at major NRTI, NNRTI, PI

and INSTI codons using Stanford University's HIVdb algorithm (https://hivdb.stanford.edu/). The sequence data were submitted to GenBank and obtained accession numbers ON337215-ON337345 for protease and reverse transcriptase (PRRT) sequences and ON337346-ON337476 for integrase (IN) sequences.

## Data analysis

We performed statistical analysis using IBM SPSS Statistics for Windows, version 23.0 Armonk, NY: IBM Corp. The descriptive statistics summarize the participants' baseline demographic, clinical characteristics, VS and DRMs as primary outcomes. Numerical variables were not normally distributed; therefore, we used frequencies for categorical variables and median (IQR). Multivariable Logistic regression model performed the association between participant characteristics and VS. Furthermore, due to the small sample size of the study participants with high VL $\geq$1000copes/mL, we used Fisher's exact test to assess the association between DRM and the participants' characteristics. The statistical significance was set at $p < 0.05$.

## Ethical consideration

Ethical approval to conduct this study was obtained from the Research Ethics Committee of the Muhimbili University of Health and Allied Sciences (MUHAS-REC-11-2020-422) and the National Institute for Medical Research (NIMR) (NIMR/HQ/R.8a/Vol. IX/I 3432). All participants provided written informed consent and assent accordingly before recruitment into the study. Participants genotyping results guided the recommendation of adherence interventions or switching ART for those with clinically significant DRMs to the attending clinicians at their respective facilities.

## Results

### Study participant characteristics

Of the 578 recruited AYA, data of 570 AYA with valid VL results was analyzed (**Table 1**). The median age was 13.0 years (Interquartile Range (IQR)11.0–14.0). Adolescents aged 10–19 years were the majority, 92.4% (527/570), and more than half were females, 55.1% (314/570). Primary education and below were attained by 92.1% (525/570). At the time of enrollment, INSTI based ART regimen was the most frequently used by 76.8% (438/570) of participants.

### Viral suppression

The prevalence of VS among AYA receiving ART between 9-15months and $\geq$36 months that was assessed as early and late time points were 91.5% and 88.2%, respectively. Overall, about 88.0% (464//527) of adolescents aged 10–19 years achieved VS, while 97.7% (42/43) of young adults (20–24 years) achieved VS. Furthermore, we observed a higher proportion of females with VS (89.2%) and participants on INSTI based regimen 90.9%.

### Factors associated with non-viral suppression

Table 2 demonstrates factors associated with non-viral suppression in the multivariable logistic regression model. In bivariate analysis, AYA who were currently on PI-based regimen were twice as likely to not achieve VS with a crude odds ratio (cOR = 2.44, 95% CI = 1.27–4.68, *p value* .007) compared to those on INSTI regimen. After adjusting for other covariates, the PI regimen remained statistically significantly associated with non-viral suppression, adjusted odd ratio (aOR = 2.09, 95% CI = 1.03–4.24, *p value* .041) in the multivariable model.

**Table 1. Viral suppression status by demographic and clinical characteristics of HIV-infected adolescents (10–19 years) and young adults (20-24years) receiving antiretroviral therapy (n = 570).**

| | | HIV viral load testing results | | |
| --- | --- | --- | --- | --- |
| Variable | | Suppressed | Non-suppressed | Total |
| | | n (%) | n (%) | |
| Age group (years) | | | | |
| | 10–19 | 464 (88.0) | 63 (12.0) | 527 |
| | 20–24 | 42 (97.7) | 1 (2.3) | 43 |
| Median age in years (IQR) | | 13 (11, 14) | 13 (11, 14) | 570 |
| Sex | | | | |
| | Female | 280 (89.2) | 34 (10.8) | 314 |
| | Male | 226 (88.3) | 30 (11.7) | 256 |
| Education | | | | |
| | Primary and below | 463 (88.2) | 62 (11.8) | 525 |
| | Secondary and above | 43 (95.6) | 2 (4.4) | 45 |
| Median age at ART initiation (IQR) (years) | | 7 (3, 10) | 7 (3, 9) | 518 |
| ART regimen | | | | |
| | NNRTI based | 13 (92.9) | 1 (7.1) | 14 |
| | PI based | 52 (77.6) | 15 (22.4) | 67 |
| | INSTI based | 398 (90.9) | 40 (9.1) | 438 |
| | Missing ART regimen | | | 51 |

## Prevalence of acquired drug resistance

The study found that 71.9% (46/64) of participants with VL≥1000 copies/mL had at least one DRM. Most mutations selected the NNRTIs class at 67.7%, followed by NRTIs at 43.5%, PI major at 3.2% and INSTI major at 6.3% (Fig 2). Dual-class resistance occurred with

**Table 2. Factors associated with non-viral suppression among adolescents and young adults receiving anti-retroviral therapy (n = 506).**

| | | Univariate analysis | | | Multivariate analysis | | |
| --- | --- | --- | --- | --- | --- | --- | --- |
| Variable | Category | cOR | 95% CI | P—value | aOR | 95% CI | P—value |
| Age group (years) | 10–19 | 5.70 | 0.77–42.16 | 0.088 | 3.38 | 0.38–30.23 | 0.276 |
| | 20–24 | Ref | | | | | |
| Sex | Male | 1.09 | 0.65–1.84 | 0.738 | 1.47 | 0.81–2.68 | 0.205 |
| | Female | Ref | | | | | |
| Education | Primary and below | 1.56 | 0.72–3.38 | 0.264 | 1.04 | 0.43–2.52 | 0.935 |
| | Secondary | Ref | | | | | |
| ART regimen | NNRTI | 0.75 | 0.10–5.88 | 0.784 | 0.95 | 0.12–7.65 | 0.962 |
| | PI | 2.44 | 1.27–4.68 | 0.007 | 2.09 | 1.03–4.24 | **0.041** |
| | INSTI | Ref | | | | | |
| Duration on ART | ≥ 36 | 1.27 | 0.52–3.10 | 0.605 | 0.79 | 0.28–2.22 | 0.658 |
| | 16–35 | 0.83 | 0.20–3.48 | 0.794 | 0.68 | 0.15–2.98 | 0.605 |
| | ≤ 15 | Ref | | | | | |
| Number of ART changes | ≥ 4 | 1.73 | 0.95–3.14 | 0.072 | 1.33 | 0.67–2.65 | 0.414 |
| | < 4 | Ref | | | | | |
| Ever had ART side effect | Yes | 2.84 | 0.99–8.05 | 0.050 | 1.64 | 0.44–6.09 | 0.461 |
| | No | Ref | | | | | |

Key: cOR: crude Odds Ratio, aOR: adjusted Odds Ratio

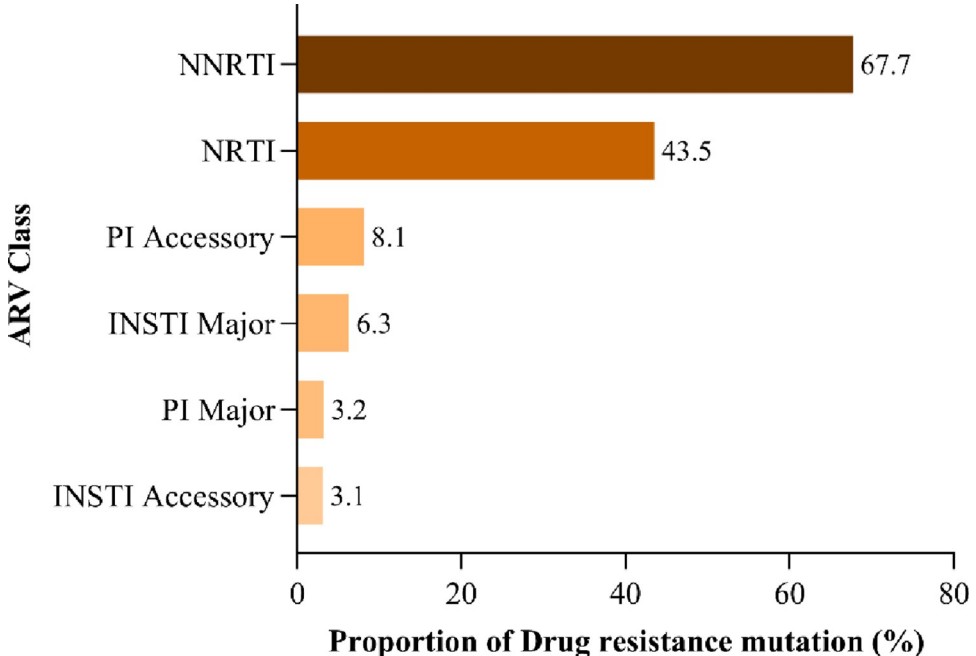

**Fig 2. The proportion of drug-resistant mutations selected by antiretroviral drug classes among adolescents and young adults.**

NRTIs plus NNRTIs at 41.9%, NRTI plus PI/r at 6.5%, NRTI and INSTI at 6.5% and less than 2% with PI and INSTI. The multi-class resistance NRTI, PI and INSTI was detected at < 2% (Fig 3).

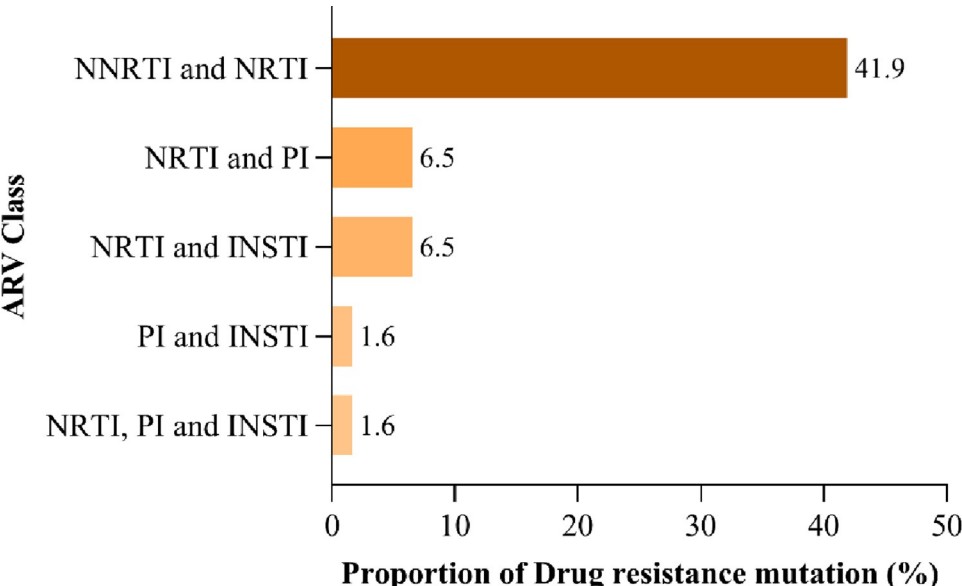

**Fig 3. The proportion of drug-resistance mutations selected by ARV classes showing dual and triple class resistance among adolescents and young adults.**

## Clinically relevant drug-resistant mutations among adolescents and young adults

The most frequent mutations in the NNRTI class were K103N (42.9%), which occurred among participants on ART between 9–15 (early time point) and > 36 months (late time point). We detected V179E and E138A in 28.6% during the early time point (9–15) months. In addition, mutations Y181C/Y and Y318F occurred in 14.3% of the participants on ART between 16–35 months and above. About 20.8% of participants with an ART duration of > 36 months had G190A and A98G (Fig 4).

The most common mutations in the NRTI class were detected at early point (9–15 months), including M184V, T215Y and L210W/L. DRMs detected at late timepoint (>36months) include M41L, D67N, K70R, T215Y/F and K219E. The most frequent DRM was M184V (42.9%) followed by the thymidine associated mutations (TAMs) M41L (28.6%), T215Y/F (28.6%), L210W/L (14.3%), K70R 14.6% and D67N (14.6%) (Fig 5). We did not find the non-thymidine associated mutation K65R, even though most participants were on a tenofovir based regimen. Low levels of PI resistance mutations were observed among AYA receiving a PI-based ART. The PI major mutation L89V/T was found in 14.3% of participants, and they were on ART between 16–35 months (Fig 6). INSTI major DRMs included G118R, E138K, and T66A; these occurred in 14.3% of the study participants with ADR (Fig 7). Overall, we observed that the prevalence of DRMs increased with time on ART.

## HIV subtypes

The maximum likelihood phylogenetic tree (Fig 8) illustrates the distribution of HIV-1 sub-types among AYA with high viremia with successful genotyped viral sequences (N = 64). Our phylogenetic analysis indicates that the predominant HIV-1 subtypes among AYA with high viremia is subtype C (n = 28), followed by subtype A1(n = 19). Other subtypes were recombinant A1, C (n = 7); recombinant A1, D (n = 4); subtype D (n = 3); recombinant D, C (n = 2); CRF_10_CD (n = 2) and CRF_35_AD (n = 1). Overall, 45.6% of AYA harboring at least one DRM had HIV-1 subtype C (n = 21/46) and 26.1% had subtype A1 (n = 12/46).

## Factors associated with acquired drug resistance mutations among adolescents and young adults receiving anti-retroviral therapy

In this study, we analysed age, gender, initial ART regimen, duration on ART, frequency of ART regimen change, ever experience ART side effects, latest CD4 T-cell count, disclosure of HIV status and ART adherence. We study found that male gender had a higher proportion of ADR (83.3%), although this was not statistically significant (Table 3). In addition, the initial VL result (6 months after ART initiation) of ≥1000copies/mL was associated with detecting DRMs (*p value* = .019).

## Discussion

The current study offers three key findings among AYA; first, the overall rate of VS was below the third UNAIDS 90 target by the year 2020. Second, there is a high proportion of HIV-DRM (71.9%) among AYA with viral load1000copies/mL. Third, the first HVL test result of ≥1000copies/mL after ART initiation was a significant risk factor associated with the emergence of ADR.

VS and ADR are not uniform across diverse regions and populations; our findings show a higher level of VS than in other African settings, where VS among adolescents ranges between 48.4% and 79% [10, 30]. Similarly, Zimbabwe had a higher level (89.8%), close to achieving the

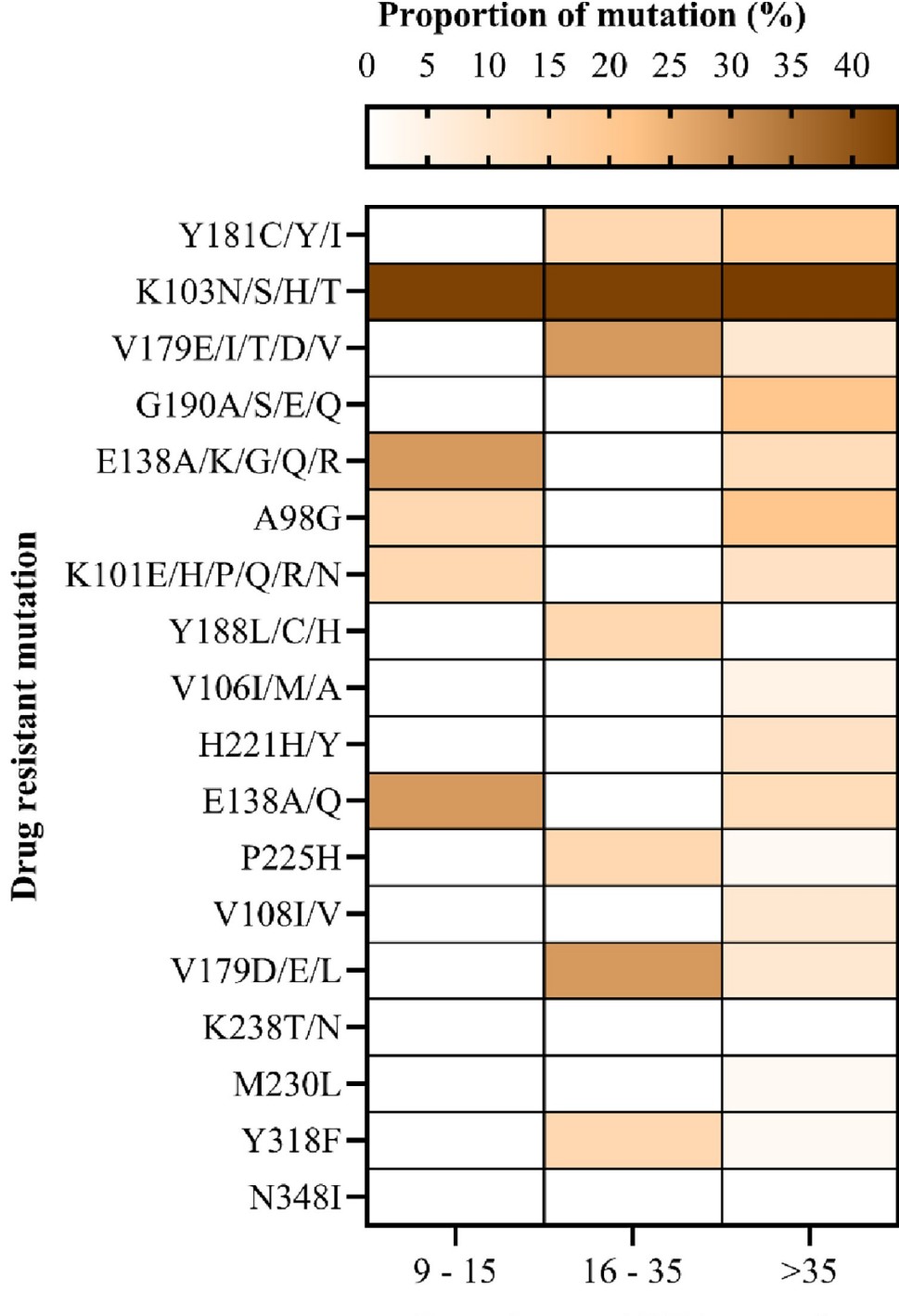

**Fig 4. The proportion of drug-resistant mutations selected by non-nucleoside reverse transcriptase inhibitors among adolescents and young adults.**

2020 UNAIDS target [31]. However, in our study, VS among AYA receiving ART for ≥36 months was comparatively lower (88.2%) than those receiving ART for 12 ± 3 months (91.5%). In contrast, in ADR surveys reported by the WHO in 2021; in Zambia, children and

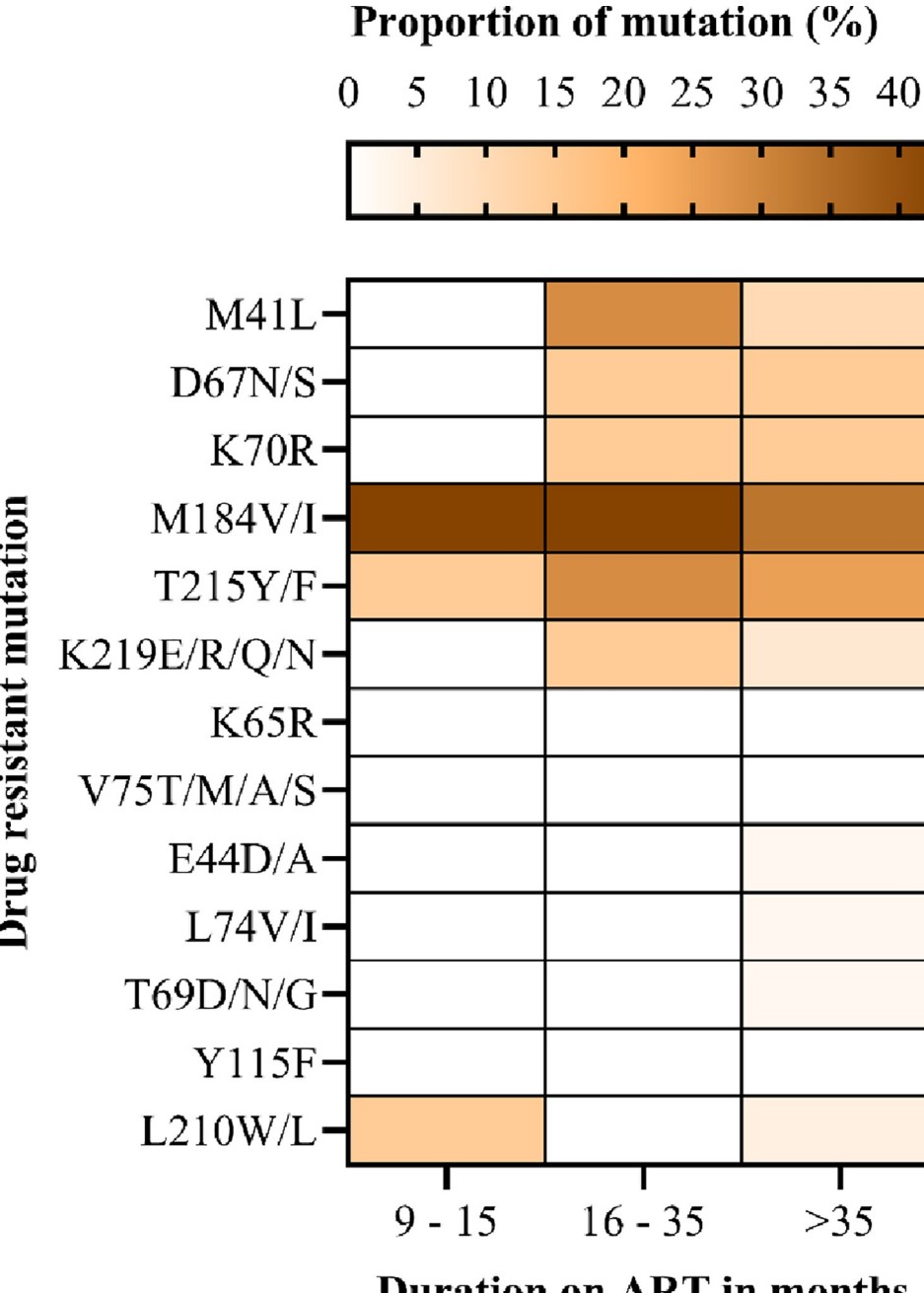

**Fig 5. The proportion of drug-resistant mutations selected by the nucleoside reverse transcriptase inhibitors among adolescents and young adults.**

adolescents had suboptimal VS at 69% during early time point assessment (receiving ART for 12 ± 3 months) and 67% at a late time point (receiving ART for ≥36 months) [32]. The difference in the levels of VS may be due to the rapid VS following the rollout of DTG-based first-line regimens at different times between 2019 and 2020 in most African countries. Indeed, during the present ADR survey in Tanzania, more than 70% of AYA were on DTG-based regimens compared to only 39% of children and adolescents in the Zambian study [32].

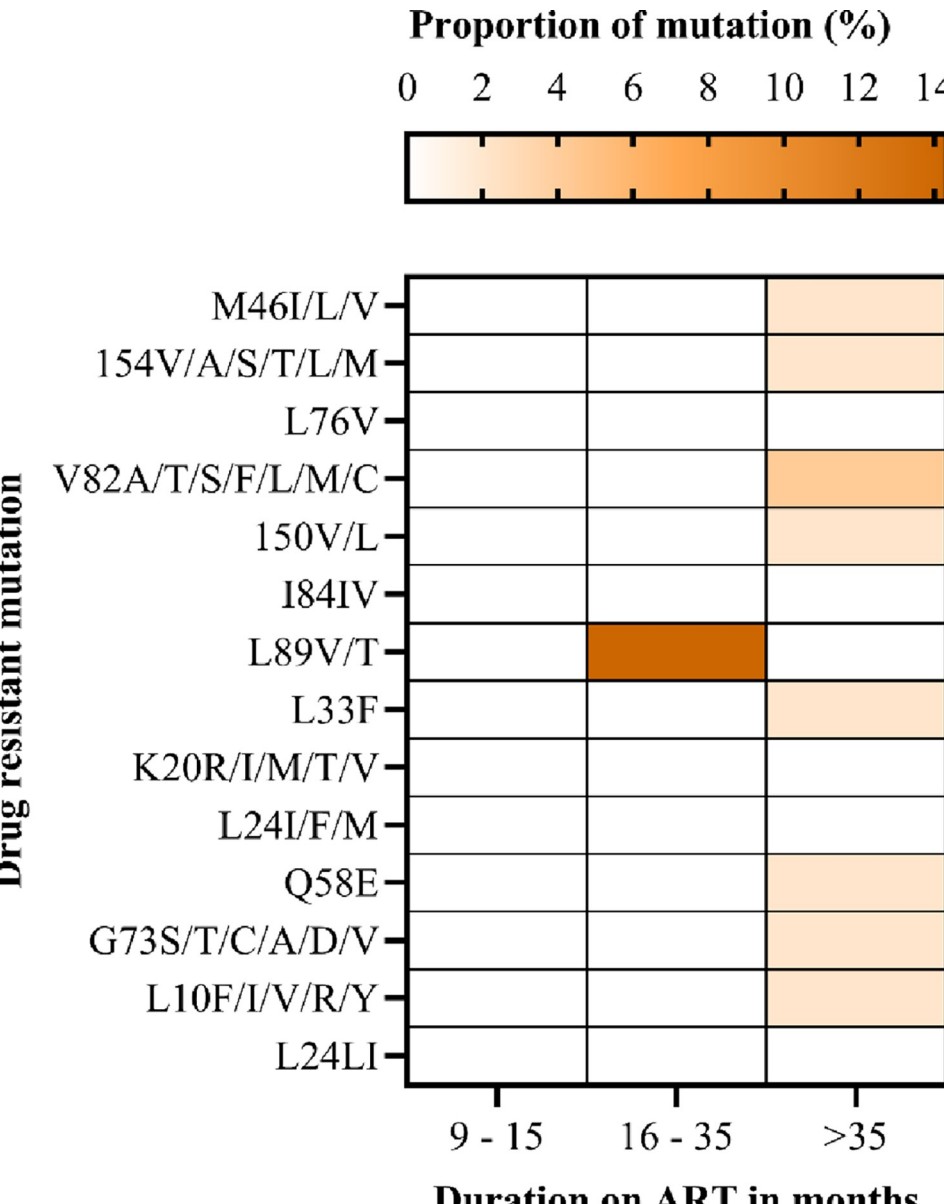

**Fig 6. The proportion of drug-resistant mutations selected by protease inhibitors among adolescents and young adults.**

Importantly, AYA who were currently on PI-based regimen were twice as likely to not achieve VS compared to those on INSTI regimen (DTG). This observation is consistent with the fact that DTG-based regimens are associated with faster VS and a higher genetic barrier to resistance [33]. Further, DTG-based regimens have shown better outcomes compared to non DTG for first generation NNRTIs and PIs [33, 34]. Therefore, moving towards the UNAIDS 95 95 95 targets by 2030, scale up of DTG based regimen in adolescents is paramount.

There are different determinants of suboptimal to non-adherence and, non-viral suppression within the age group 10–24 years. Young adolescents (10–15 years) and older adolescents (15–17 years) face adherence challenges due to developmental changes and leading to an inability to undertake the task of their HIV management [35]. Factors such as dependence on

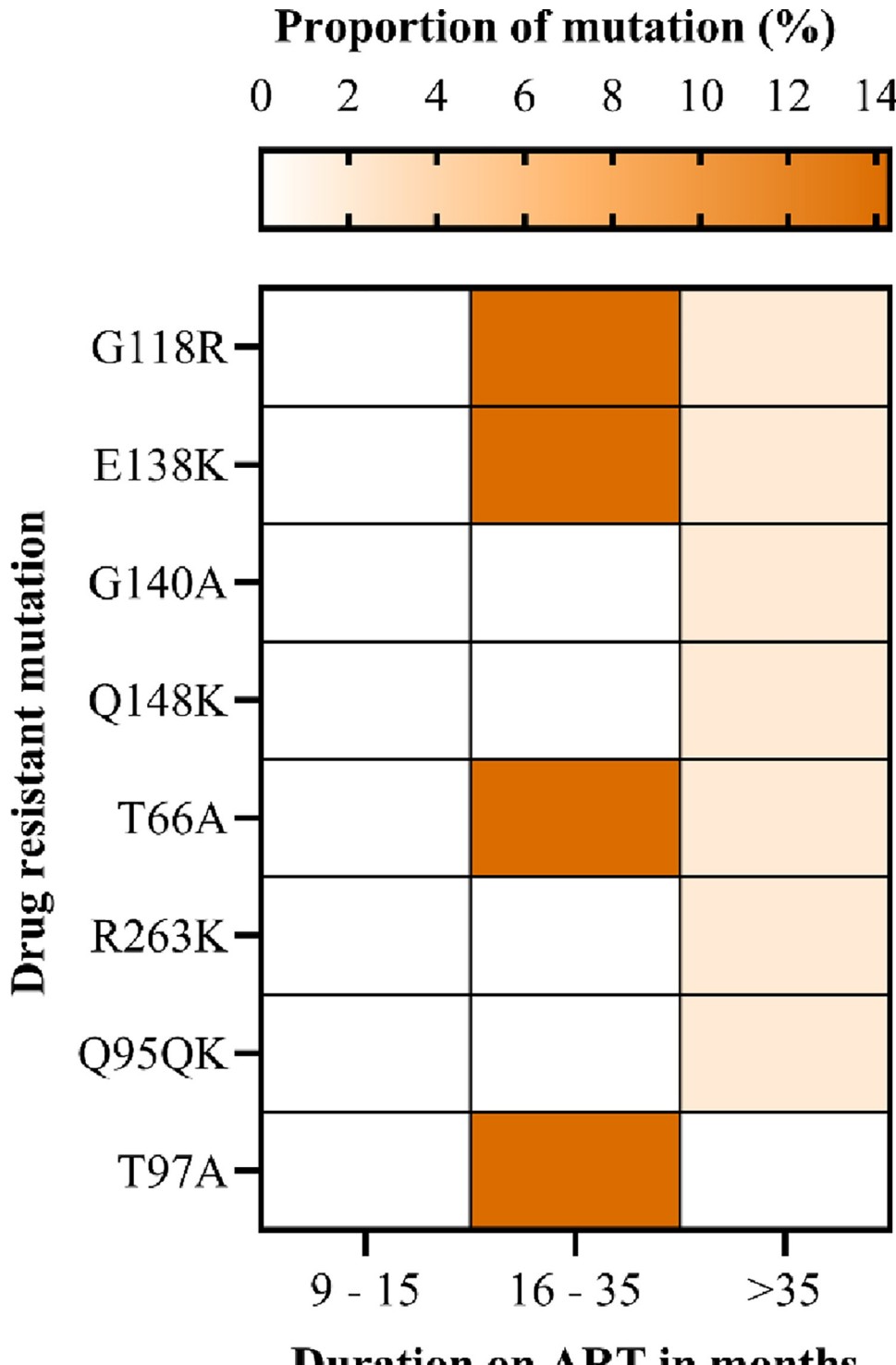

**Fig 7. The proportion of drug-resistant mutations selected by integrase strand transfer inhibitors among adolescents and young adults.**

parents, guardians, family settings for medication administration, keeping clinic appointments, disclosure of HIV-positive status, attending school, type and dosing of ARV, and

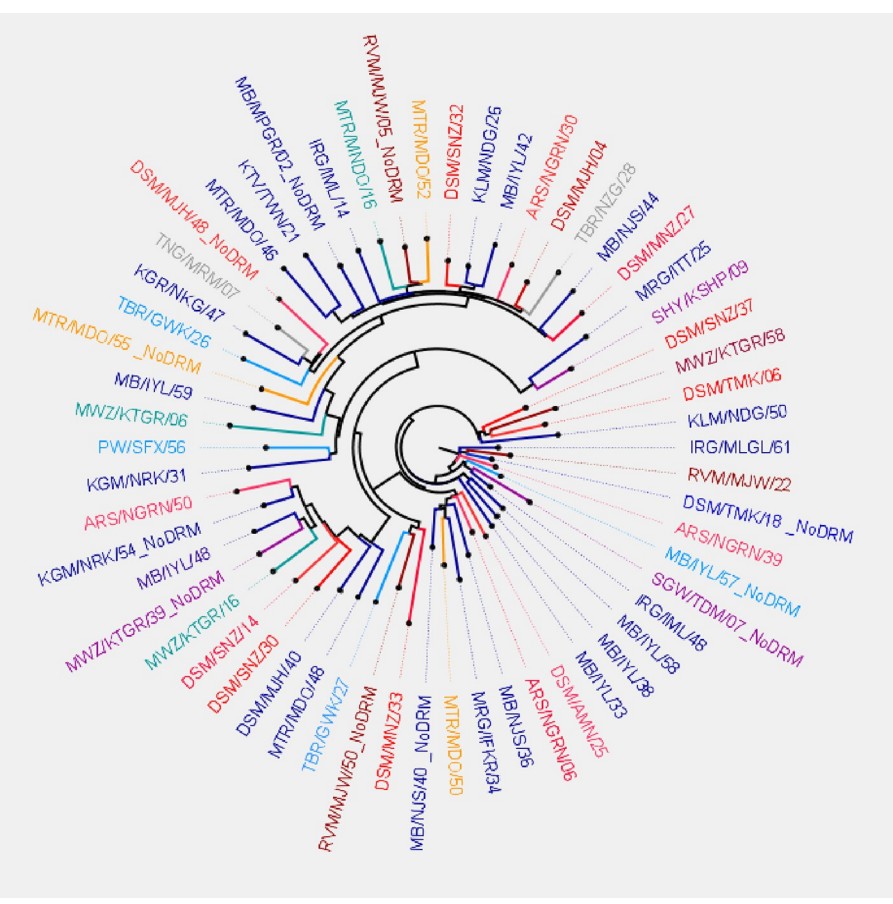

Key

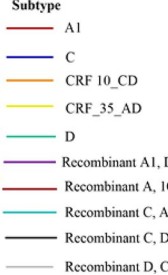

| Subtype | |
|---|---|
| —— | A1 |
| —— | C |
| —— | CRF 10_CD |
| —— | CRF_35_AD |
| —— | D |
| —— | Recombinant A1, D |
| —— | Recombinant A, 1C |
| —— | Recombinant C, A1 |
| —— | Recombinant C, D1 |
| —— | Recombinant D, C |

**Fig 8. Phylogenetic tree illustrating the distribution of HIV-1 subtypes among participants with high viremia.**

patient-provider relationship may adversely affect their adherence to ART, leading to ART failure and/ or development of drug resistance [35, 36]. Determinants of adherence and treatment failure in young adults (18–24 years) include self-stigma, non-disclosure of HIV-positive status to family members and sexual partners, age-related behaviors such as alcoholism, illegal drug abuse and transitioning from adolescence to adult HIV care [36].

In addition, existing AYA-friendly services in SSA countries, Tanzania inclusive, need to include social and behavioral factors that may influence ART adherence and health system

**Table 3. Factors associated with acquired drug resistance mutations among adolescents and young adults initiated on antiretroviral therapy attending care and treatment program in Tanzania (n = 64).**

| Variable | | Anti-retroviral drug resistance | | |
|---|---|---|---|---|
| | | Present (%) | Absent (%) | P–value |
| Age group (years) | | | | |
| | 10–14 | 39 (72.2) | 15 (27.8) | 1.000 |
| | 15–24 | 7 (70.0) | 3 (30.0) | |
| Gender | | | | |
| | Male | 25 (83.3) | 5 (16.7) | 0.093 |
| | Female | 21 (61.8) | 13 (38.2) | |
| Education | | | | |
| | None | 21 (75.0) | 7 (25.0) | 0.798 |
| | Primary | 20 (71.4) | 8 (28.6) | |
| | Secondary | 5 (62.5) | 3 (37.5) | |
| Initial ARV classes | | | | |
| | INSTI | 5 (50.0) | 5 (50.0) | 0.804 |
| | NNRTI | 40 (76.9) | 12 (23.1) | |
| | PI | 1 (50.0) | 1 (50.0) | |
| Duration on ART (months) | | | | |
| | 11–15 | 4 (57.1) | 3 (42.9) | 0.362 |
| | 16–35 | 4 (57.1) | 3 (42.9) | |
| | >35 | 38 (76.0) | 12 (24.0) | |
| Number regimen change | | | | |
| | < 4 | 17 (63.0) | 10 (37.0) | 0.260 |
| | ≥ 4 | 29 (78.4) | 8 (21.6) | |
| Ever experienced ART side effects | | | | |
| | Yes | 7 (87.5) | 1 (12.5) | 0.424 |
| | No | 39 (69.6) | 17 (30.4) | |
| Initial HIV viral load result | | | | |
| | <1000copies/ml | 12 (52.2) | 11 (47.8) | **0.019** |
| | ≥1000copies/ml | 34 (82.9) | 7 (17.1) | |
| Latest CD4 count | | | | |
| | < 350 | 9 (81.8) | 2 (18.2) | 0.714 |
| | ≥ 350 | 37 (69.8) | 16 (30.2) | |
| Disclosure* | | | | |
| | Yes | 35 (68.6) | 16 (31.4) | 0.307 |
| | No | 5 (100) | 0 (0.0) | |
| Adherence* | | | | |
| | Good | 30 (70) | 11 (26.8) | 0.741 |
| | Poor | 10 (66.7) | 5 (33.3) | |

*Disclosure of HIV status defined as disclosure of HIV positive status to the participant (adolescent or young adults) and/or to at least one family member.

*ART adherence defined as using self-report assessment by healthcare provider to be good when ≥ 95% and poor <95%.

challenges that impair VS. Importantly, routine national programmatic analysis of ART outcome data on assessing VS in young populations receiving TLD will remain crucial.

Secondly, more than 70% of AYAs with viral load ≥1000 had at least one DRM. More than half had NNRTIs mutations, followed by NRTIs mutations above 40%. These findings are

similar to other studies and surveys in SSA [11, 22, 37–40]. The high rates of HIVDR among adolescents are most likely due to prolonged NNRTIs and NRTIs exposure since childhood and ART adherence challenges. The most frequent NNRTI mutations found in the current study (K103N, Y181C/Y, G190A and A98G) are comparable to other studies, which included children <14 years and young adolescents 10–14 years [39, 41]. The low genetic barrier to resistance of most NNRTIs [13] explain the study findings. In addition, children and adolescents living with HIV may have acquired resistance mutation from their mothers (transmitted drug resistance). Moreover, NNRTIs exposure through PMTCT and during early childhood HIV treatment failed to achieve VS, leading to ADR [40]. In addition, children and adolescents living with HIV may have acquired resistance mutations from their mothers (transmitted drug resistance). Moreover, NNRTIs exposure through PMTCT and early childhood HIV treatment failed to achieve VS, leading to ADR.

The present study found a lower prevalence of NRTIs DRMs than that found in Zambia, 62% and Uganda, 50%, as reported in the WHO 2021 report [32]. The most frequent DRMs were M184V and TAMs, similar to other studies reporting adolescent data (35, 38–40). The present study detection of TAMs was probably due to prior use of Zidovudine (AZT) during PMTCT or early childhood AZT as HIV treatment. The development of TAMs is associated with prolonged treatment failure [42]; thus, we emphasize strict viral load monitoring as an intervention for early detection of treatment failure. Since AZT is the subsequent choice of NRTI in second-line ART in Tanzania, the existence of TAMs might cause subsequent treatment failure associated with new DRMs. The TAM-1 pathway that includes M41L, L210W and T215Y is described as being more common, and it confers a greater negative impact on virologic response to TDF-containing regimens [43].

Furthermore, TAMs in the present study warrant attention to our national program because TAMs are associated with cross-resistance to most NRTIs [44]. Consequently, coexisting M41L, L210W, and T215Y reduce TDF response in ART-experienced PLHIV [45]. Therefore, drug resistance testing may be required to guide the recycling of NRTIs after confirmed virologic failure. Notably, we did not find the non-thymidine associated DRM K65R among our participants. Studies have reported that, K65R rarely occurs in combination with TAMs because K65R and most TAMs exhibit bidirectional antagonism [42, 46]. In contrast, other studies report an increased frequency of K65R mutation, which Tenofovir selects as it is widely used [47, 48].

We observed clinically relevant PI DRMs at a low level below 4%, similar to other studies in SSA (21, 35, 44). Moreover, in SSA, the PI DRMs frequencies are below 10% (21, 35, 44). Moreover, in SSA, the PI DRMs frequencies are below 10% [31, 40, 41, 49], suggesting that ART failure is most likely due to suboptimal adherence. However, it was surprising that more than 85% of resistance mutations to PIs occurred in Brazilian children and adolescents who had a vertical transmission of HIV for unexplained reasons [9]. The present study detected clinically significant INSTI-DRM G118R, E138K and T66A. However, we did not have prior information on participants using first-generation INSTIs, we failed to support the sequential accumulation of these mutations [50]. Nevertheless, the possibility of amplifying pre-existing INSTI minority archived DRMs caused by the transition to DTG [50] can be the explanation for this finding. Before introducing INSTIs in Tanzania, detection of only minor resistance mutations [25] was similar to other countries in Africa [51, 52]. Accordingly, periodic ADR surveys are of paramount importance to ensure the long-lasting efficacy of tenofovir + lamivudine + dolutegravir (TLD). Furthermore, when DRMs accumulate with other significant mutations, it could reduce viral susceptibility to INSTIs.

Lastly, the initial VL result ≥1000copies/mL after ART initiation was a significant risk factor for developing DRMs, in line with other studies [41, 53]. Therefore, it underscores the

importance of early detection of VF and preventing the accumulation of DRMs. Consequently, implementing optimal VL monitoring with rapid switching to an appropriate, effective ART regimen is crucial.

Our study has some limitations; first, the population-based Sanger sequencing can miss DRMs in 30% or more of DRMs (cannot detect minority variants) [54], thus potentially under-estimating the true prevalence of DRMs. Second, we did not use the standard definition of virologic failure due to limited VL results (latest VL results prior to the survey) from routine data provided during cross-sectional data extraction. Given that the WHO recommends a defi-nition of VF to be two consecutive viral loads 3 months apart above 1,000 copies/mL with adherence counselling after the first viral load. Nevertheless, the one VL measurement of ≥1000copies/mL is the standard threshold required for detecting DRMs using DBS samples. Importantly, multiple imputations mitigated the missing participants baseline VL results (first VL result 6 months after ART initiation) in statistical analysis to determine associated factors of HIV-DRMs. Nonetheless, our findings contribute to understanding circulating HIV-DRMs among AYA as a vulnerable population disproportionately affected HIV epidemic.

## Conclusions

This first national representative ADR survey found an overall low VS below the third UNAIDs target among AYA. In addition, more than one in ten AYA with high viremia (VL≥1000copies/ml) had a high prevalence of circulating clinically relevant DRMs. The first VL result of ≥1000copies/ml after ART initiation is a significant risk factor for developing DRMs. Thus, underscoring strict VL monitoring for early identification of treatment failure and genotypic testing during ART switch to guide the choice of NRTIs backbone of two new or recycled NRTIs is recommended to improve VS. Furthermore, surveillance of DRMs select-ing for INSTI is paramount since the introduction and scale-up of DTG-based regimens in children and adolescents is ongoing in Tanzania. Therefore, attaining national and global HIV and AIDS targets calls for interventions addressing ADR among AYA.

## Supporting information

**S1 Dataset.**
(XLSX)

## Acknowledgments

The authors wish to thank the survey participants, research assistants, districts and regional AIDS Control Coordinators (DACCs and RACCs), district and regional medical officers and regional laboratory technicians (DMOs, RMOs and RLTs) and the National AIDS Control Program (NACP) team for making the first ADR survey a successful one. We also acknowl-edge the support from the CDC and WHO Tanzania, the University of California San Fran-cisco (USCF), United States for their technical support and the British Columbia Centre for Excellence in HIV/AIDS, Canada, for performing HIVDR genotyping.

## Author Contributions

**Conceptualization:** Joan Rugemalila, Doreen Kamori, Mucho Mizinduko, Amon Sabasaba, Davis Amani, Erick Mboya, George Ruhago, Veryeh Sambu, George Mgomella, Boniface Jullu, Werner Maokola, Prosper Njau, Takamasa Ueno, Tumaini Nagu, Bruno Sunguya, Said Aboud.

**Data curation:** Doreen Kamori, Peter Kunambi, Mucho Mizinduko, Amon Sabasaba, Salim Masoud, Boniface Jullu.

**Formal analysis:** Joan Rugemalila, Peter Kunambi, Mucho Mizinduko, Amon Sabasaba.

**Funding acquisition:** Veryeh Sambu, Werner Maokola, Prosper Njau, Beatrice Mutayoba, Tumaini Nagu, Bruno Sunguya, Said Aboud.

**Investigation:** Joan Rugemalila, Doreen Kamori, Salim Masoud, Sabina Mugusi, Rita Mutagonda, Linda Mlunde, Davis Amani, Erick Mboya, Macdonald Mahiti, George Ruhago, Jeremiah Mushi, Veryeh Sambu, George Mgomella, Werner Maokola, Prosper Njau, Bruno Sunguya, Said Aboud.

**Methodology:** Doreen Kamori, Peter Kunambi, Mucho Mizinduko, Amon Sabasaba, Salim Masoud, Frank Msafiri, Boniface Jullu, Werner Maokola, Godfrey Barabona, Takamasa Ueno, Tumaini Nagu, Said Aboud.

**Project administration:** Joan Rugemalila, Doreen Kamori, Mucho Mizinduko, Frank Msafiri, Sabina Mugusi, Rita Mutagonda, Linda Mlunde, Davis Amani, Erick Mboya, Macdonald Mahiti, Jeremiah Mushi, Veryeh Sambu, Boniface Jullu, Werner Maokola, Prosper Njau, Beatrice Mutayoba, Andrea Pembe, Bruno Sunguya.

**Resources:** Werner Maokola, Said Aboud.

**Supervision:** Joan Rugemalila, Doreen Kamori, Mucho Mizinduko, Salim Masoud, Frank Msafiri, Sabina Mugusi, Rita Mutagonda, Linda Mlunde, Davis Amani, Erick Mboya, Macdonald Mahiti, George Ruhago, Jeremiah Mushi, Veryeh Sambu, Boniface Jullu, Werner Maokola, Prosper Njau, Tumaini Nagu, Bruno Sunguya.

**Writing – original draft:** Joan Rugemalila, Rita Mutagonda, George Mgomella, Boniface Jullu, Werner Maokola.

**Writing – review & editing:** Doreen Kamori, Frank Msafiri, Godfrey Barabona, Takamasa Ueno, Tumaini Nagu, Bruno Sunguya, Said Aboud.

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
