## [Decision Letter · Decision Letter 0]

31 Oct 2022

PONE-D-22-26318HIV-1 Virologic Response, Patterns of Drug Resistance Mutations and Its Associated Factors Among Adolescents and Young Adults in The Context of Rollout of Dolutegravir: A Cross-Sectional Study in TanzaniaPLOS ONE

Dear Dr. Rugemalila,

Thank you for submitting your manuscript to PLOS ONE. After careful consideration, we feel that it has merit but does not fully meet PLOS ONE’s publication criteria as it currently stands. Therefore, we invite you to submit a revised version of the manuscript that addresses the points raised during the review process.

We look forward to receiving your revised manuscript.

Kind regards,

Jason T. Blackard, PhD

Academic Editor

PLOS ONE

Journal Requirements:

"The HIV Global Fund for Malaria, Tuberculosis and HIV/AIDS in Tanzania supported the ADR survey through the Ministry of Health Community Development Gender Elderly and Children. Grant name: TZA-H-MOF and grant number 1573. JR received a scholarship from the Swedish International Development Agency (SIDA) for the postgraduate project study."

Additional Editor Comments:

This is a cross-sectional study of HIV drug resistance in adolescents and young adults in Tanzania.

PCR primers and their location relative to HIV should be included in the methods.

Line 177 and elsewhere:  “twice more unlikely” should be changed to twice as likely to not achieve VS

HIV subtypes should be included in a phylogenetic tree and incorporated into the univariate/multivariate analyses when possible.

Reviewers' comments:

Reviewer's Responses to Questions

**Comments to the Author**

1. Is the manuscript technically sound, and do the data support the conclusions?

Reviewer #1: Yes

Reviewer #2: Partly

2. Has the statistical analysis been performed appropriately and rigorously? 

Reviewer #1: Yes

Reviewer #2: Yes

3. Have the authors made all data underlying the findings in their manuscript fully available?

Reviewer #1: Yes

Reviewer #2: Yes

4. Is the manuscript presented in an intelligible fashion and written in standard English?

Reviewer #1: Yes

Reviewer #2: No

5. Review Comments to the Author

Reviewer #1: This manuscript describes drug resistance in adolescent youth, a key population that is difficult to manage in HIV treatment programs. Tanzania like many other African HIV treatment programs transitioned to dolutegravir based regimens with the potential for reduced drug resistance. Therefore, this study is important in documenting rates of drug resistance in study subjects largely on DTG ART regimens.

It is important for the authors to indicate the date that the DTG regimen change went into effect and to also discuss this in their results where non-DTG (or PI based regimens) were more associated with virologic failure. This is a critical piece of the discussion of results that is missing.

There are limitations to the study design including the assessment of only one VL measurement, as described. The authors should perform a sensitivity analysis to determine the impact of this on their analysis.

Another major limitation of the study is the use of DBS for DRM testing which might have lower sensitivity, this should be discussed. It is also possible that rates of VS determined on DBS may differ from published literature on VS that used plasma samples. Therefore, this should be detailed in the discussion of other VS studies.

The AYA study population spans 10-24 years of age, and it is likely that there are different determinants of treatment and adherence failure within that broad age range that are not discussed.

The authors state that females had higher levels of viral suppression compared to men, if so, this should be supported with a statistical test.

Minor points:

The AYA acronym should be defined the first time it is used.

Line 168 - there is an error in young adult age range (20-14 yrs) this should be 20-24?

Line 176 - “AYA who were currently on PI-based regimen were twice more unlikely to achieve VS…” Please correct the wording here “ 2x less likely to achieve VS”

Reviewer #2: This is a cross-sectional study of 570 adolescents and young adults living with HIV in Tanzania, performed to examine viral suppression and the development of drug resistance mutations (DRMs). The conclusions that higher viral load (lack of suppression) is associated with DRMs is certainly not unexpected. The findings from this small sample are largely descriptive. Data from only 64 samples were suitable for detection of DRMs, limiting the weight of the conclusions regarding the observed DRMs. There are some aspects of the study that are confusing and require further explanation. Overall the conclusions are fairly self-evident from this small survey.

Major critique:

1. The definition of viral suppression is not clear. Was this undetectable virus? What was the copy number cutoff for undetectable virus according to the methodology used? How many samples were obtained over what period of time in order to determine viral suppression in an individual subject? There is mention of “early” and “late” timepoints (line 202), presumably this means multiple samples were obtained from the subjects in this study rather than a purely cross-sectional, single timepoint value. The authors should provide complete definitions and clarify the study design as relates to viral load measurement.

2. Figure 1 shows 570 enrolled subjects with valid viral load results, yet all 570 had VL<1000. Where do the 64 VL>1000 used for sequencing come from? Presumably these 64 samples all must come from the 570 enrolled participants. Does that mean that all 570 had VL<1000 at some timepoints, while 64 of these same subjects had VL>1000 at one (or more) timepoint? Related to critique #1, does this mean that 64/570 or 11% of subjects failed to suppress virus during the duration of the study?

3. DRMs were associated with VL>1000. Does that mean a single value of VL>1000, or does that mean VL>1000 over an extended period of time, multiple measurements?

Minor: The writing is generally very interpretable, but requires editing for simple grammatical errors throughout the title and text. I did not attempt to outline the many sentences involved. An example: for the title: “HIV-1 Virologic Response, Patterns of Drug Resistance Mutations and Associated Factors…”

6. PLOS authors have the option to publish the peer review history of their article (what does this mean?). If published, this will include your full peer review and any attached files.

Reviewer #1: No

Reviewer #2: No

---

## [Author Response · Author response to Decision Letter 0]

6 Jan 2023

We express our sincere appreciation to the editor and reviewers for taking the time to review our paper and to provide very good comments. Your valuable comments have led to improvement in the current version. We have carefully considered each comment and, as a result, present our best responses to each of them. Our responses are provided in a point-to- point manner below. All changes in the revised manuscript are highlighted in yellow. 

We have addressed the editorial point as follows:

Point 1: PCR primers and their location relative to HIV should be included in the methods.

Response: We have added the following statement in relation to PCR primers “Sets of routine outer and inner forward primers ; 5’PROT1:TAATTTTTAGGGAAGATCTGGCCTTCC;: AGTAGGACCTACACCTGTCA; 5’PROT2: TCAGAGCAGACCAGAGCCAACAGCCCC; and A35: TTGGTTGCACTT TAAATTTTCCCATTAGTCCTATT‐and reverse primers 3’PROT1:GCAAATACTGGAGTATTGTATGGATTTTCAGG; MJ4:CTGTTAGTGCTT TGGTTCCTCT; 3’PROT2: AATGCTTTTATTTTTTCTTCTGTCAATGGC; and 3’ NE135: CCTACTAACTTCTGTATGTCATTGACAGTCCAGCT for PR-RT region were used for PCR amplification and sequencing. Whereas, for IN region the forward primers INPS1: TAGTAGCCA GCTGTGATAAATGTC and INPS3: GAAGCCATGCATGG CAAG; and reverse primers INPR8: TTCCATGTTCTAATCCTCATC CTG and INPR9: ATCCTCATCCTGTCTACT TGCC were used. A set of inner MJ3 PCR primers for PR-RT and IN regions were used for Sanger sequencing using Applied Biosystems 3730XL DNA Analyser according to manufacturer’s instructions”. Please refer to lines 158-174.

Point 2: Line 177 and elsewhere: “twice more unlikely” should be changed to twice as likely to not achieve VS

Response: twice more unlikely has been changed to read “twice as likely to not achieve VS lines 46 and 231-233. 

Point 3: HIV subtypes should be included in a phylogenetic tree and incorporated into the univariate/multivariate analyses when possible.

Response: We have included a phylogenetic tree as illustrated by figure 4. We have indicated (bold) sequences of AYA with drug resistance mutations (n=46). Our phylogenetic analysis indicates that the predominant HIV-1 subtypes among AYA with high viremia is subtype C (n=28), followed by subtype A1(n=19). Please refer to lines 292 to 301.

Furthermore, we would like to inform reviewers that some of the authors of this manuscript have performed a detailed Phylogenetic analysis using the national HIV drug resistance surveillance data and a manuscript is under development entitled “HIV subtypes and acquired drug resistance: Findings from national HIV drug resistance surveillance in Tanzania. 

Reviewer no 1

Point 1: It is important for the authors to indicate the date that the DTG regimen change went into effect and to also discuss this in their results where non-DTG (or PI based regimens) were more associated with virologic failure. This is a critical piece of the discussion of results that is missing.

Response: We thank the reviewer for the comment. We have included a date when DTG transition went into effect in Tanzania. Please refer line 101 and 102. We have also discussed that non-DTG regimens (PI based) were more associated with high viral load ≥1000 copies/mL compared to those on DTG regimens. Kindly refer to lines 342-347 in the discussion section. 

Point 2: There are limitations to the study design including the assessment of only one VL measurement, as described. The authors should perform a sensitivity analysis to determine the impact of this on their analysis.

Response: We acknowledge this valid comment by the reviewer. To address this, the limitation statement has been rephrased for clarity: “we did not use the standard definition of virologic failure due to limited latest VL results prior to the survey from routine data provided during cross-sectional data extraction. Given that, the WHO recommends a definition of VF to be two consecutive viral loads 3 months apart above 1,000 copies/mL with adherence counselling after the first viral load. Nevertheless, the one VL measurement of ≥1000copies/mL is the standard threshold required for detecting DRMs using DBS samples.

This description has been added in lines 434 to 441. 

Point 3: Another major limitation of the study is the use of DBS for DRM testing which might have lower sensitivity, this should be discussed. It is also possible that rates of VS determined on DBS may differ from published literature on VS that used plasma samples. Therefore, this should be detailed in the discussion of other VS studies. 

Response: We have addressed this valid comment by including a sentence in the methods section “HIVDR genotyping was primarily done from DBS samples and plasma was used as a backup for samples that were not successfully sequenced from DBS. Please refer to lines 149 to155. Additionally, we would like to inform that the methodological limitations of using DBS for DRM testing have already been discussed our published protocol for the national drug resistance surveillance with reference 27 in our manuscript.

Rugemalila J, Kamori D, Maokola W, et al. Acquired HIV drug resistance among children and adults receiving antiretroviral therapy in Tanzania: a national representative survey protocol. BMJ Open 2021;11: e054021. doi:10.1136/ bmjopen-2021-054021

Regarding the rates of VS determined on DBS may differ from published literature on VS that used plasma samples. We have pointed out that during the national ADR surveillance, plasma samples were used to determine VS using existing VL testing platform in Tanzania and DBS samples shipped to the WHO accredited laboratory at the British Columbia Centre of Excellence in HIV/AIDS in Canada for genotyping. Please refer to lines 139-141 and 151 to 153.

Point 4: The AYA study population spans 10-24 years of age, and it is likely that there are different determinants of treatment and adherence failure within that broad age range that are not discussed. 

Response: We have added a paragraph describing different determinants of treatment and adherence failure in the discussion section to address this important comment from line 350 to 362. 

Point 5: The authors state that females had higher levels of viral suppression compared to men, if so, this should be supported with a statistical test.

Response: The statement in question has been modified in describing baseline characteristics of study participants to improve clarity “we observed a higher proportion of females with VS (89.2%)”. Please refer to lines 221 to 222.

Point 6: Minor points:

The AYA acronym should be defined the first time it is used.

Line 168 - there is an error in young adult age range (20-14 yrs) this should be 20-24?

Line 176 - “AYA who were currently on PI-based regimen were twice more unlikely to achieve VS…” Please correct the wording here “2x less likely to achieve VS”

Responses: 

• We have defined AYA the first time it is used on line 30-31

• an error in young adult age range (20-14 yrs) has been changed to 20-24years line 220

• The wording “twice less likely to achieve VS” has been corrected to read “were twice as likely to not achieve VS” lines 46 in abstract and lines 231 in results section.

Reviewer no 2

Point 1: This is a cross-sectional study of 570 adolescents and young adults living with HIV in Tanzania, performed to examine viral suppression and the development of drug resistance mutations (DRMs). The conclusions that higher viral load (lack of suppression) is associated with DRMs is certainly not unexpected. The findings from this small sample are largely descriptive. Data from only 64 samples were suitable for detection of DRMs, limiting the weight of the conclusions regarding the observed DRMs. There are some aspects of the study that are confusing and require further explanation. Overall the conclusions are fairly self-evident from this small survey.

Response: the conclusion has been revised to improve clarity in lines 449-452.

Point 2: Major critique

The definition of viral suppression is not clear. Was this undetectable virus? What was the copy number cut off for undetectable virus according to the methodology used? How many samples were obtained over what period of time in order to determine viral suppression in an individual subject? There is mention of “early” and “late” timepoints (line 202), presumably this means multiple samples were obtained from the subjects in this study rather than a purely cross-sectional, single timepoint value. The authors should provide complete definitions and clarify the study design as relates to viral load measurement.

Responses: We thank the reviewer for these comments.

• Authors did not use undetectable viral load levels to define VS. We have included a definition of viral suppression as viral load <1000copies/mL. Please refer to line 141.

• The authors acknowledge the comment about early and late time points VS. To address this, we have improved our description and clarify the terms “early” and “late” timepoints VS according to the WHO guidance. We have added a sentence which reads “At the time of survey, early time point assessment for VS included participants receiving ART for 9-15 months and late time point among those on ART for more than 36 months as per the WHO guidance. Kindly refer to lines 136 to139.

• We obtained one viral load measurement in order to determine VS in an individual subject in a cross-sectional design described in study procedures section, please refer to lines 139 to 141.

Point 3: Figure 1 shows 570 enrolled subjects with valid viral load results, yet all 570 had VL<1000. Where do the 64 VL>1000 used for sequencing come from? Presumably these 64 samples all must come from the 570 enrolled participants. Does that mean that all 570 had VL<1000 at some timepoints, while 64 of these same subjects had VL>1000 at one (or more) timepoint? Related to critique #1, does this mean that 64/570 or 11% of subjects failed to suppress virus during the duration of the study?

Response: We thank you for pointing out the errors in figure 1. We have made the following corrections: out of 578 enrolled participants, 8 were excluded in the analysis due to invalid VL results. Out of 570 participants with a valid VL result, 64 had high viremia (one time point VL ≥1000 copies/mL) and 506 had VS (one time point VL <1000copies/mL. The 64 samples out of 570 (11.2%) came from one VL measurement during the cross-sectional survey and these samples were subjected to genotype testing. The revised figure 1 described in line 183 is presented on a separate document of figures for this manuscript.

Point 4: DRMs were associated with VL>1000. Does that mean a single value of VL>1000, or does that mean VL>1000 over an extended period of time, multiple measurements?

Response: We clarify that a single VL≥1000copies/mL was used as a threshold to detect DRMs at the time of survey. Please refer to lines152-153. During analysis to determine factors associated with DRMs, we found that the initial VL≥1000copies/mL (the first VL test at 6 months after ART initiation) was associated with development of DRMs. Please refer to lines 310 to 311. We would like to add that “the initial VL and other VL testing participants’ data prior to the survey was obtained from routine patient’s records in the electronic patient monitoring system from each study site and, data abstraction was done during the survey. 

Point 5: Minor: The writing is generally very interpretable, but requires editing for simple grammatical errors throughout the title and text. I did not attempt to outline the many sentences involved. An example: for the title: “HIV-1 Virologic Response, Patterns of Drug Resistance Mutations and Associated Factors…”

Response: We acknowledge this comment and authors have made improvements in grammar throughout the manuscript. The title has been revised to read “HIV-1 Virologic Response, Patterns of Drug Resistance Mutations and Correlates Among Adolescents and Young Adults: A Cross-Sectional Study in Tanzania”

As a corresponding author, I confirm that all the authors have equally contributed to revise the comments and have agreed to resubmission. We hope that, the revised version is now suitable for publication and we look forward to hearing from you soon.

---

## [Decision Letter · Decision Letter 1]

26 Jan 2023

HIV-1 virologic response, patterns of drug resistance mutations and correlates among adolescents and young adults: a cross sectional study in Tanzania

PONE-D-22-26318R1

Dear Dr. Rugemalila,

We’re pleased to inform you that your manuscript has been judged scientifically suitable for publication and will be formally accepted for publication once it meets all outstanding technical requirements.

Kind regards,

Jason T. Blackard, PhD

Academic Editor

PLOS ONE

Additional Editor Comments (optional):

None

Reviewers' comments:

Reviewer's Responses to Questions

**Comments to the Author**

1. If the authors have adequately addressed your comments raised in a previous round of review and you feel that this manuscript is now acceptable for publication, you may indicate that here to bypass the “Comments to the Author” section, enter your conflict of interest statement in the “Confidential to Editor” section, and submit your "Accept" recommendation.

Reviewer #1: All comments have been addressed

Reviewer #2: All comments have been addressed

2. Is the manuscript technically sound, and do the data support the conclusions?

Reviewer #1: (No Response)

Reviewer #2: Yes

3. Has the statistical analysis been performed appropriately and rigorously? 

Reviewer #1: (No Response)

Reviewer #2: Yes

4. Have the authors made all data underlying the findings in their manuscript fully available?

Reviewer #1: (No Response)

Reviewer #2: Yes

5. Is the manuscript presented in an intelligible fashion and written in standard English?

Reviewer #1: (No Response)

Reviewer #2: Yes

6. Review Comments to the Author

Reviewer #1: Major concerns have been addressed with revised text and figures. Authors should ensure that they have addressed comments on small sample size.

Reviewer #2: The authors have responded appropriately to the comments, and the writing is stronger. No more concerns, just a minor typo: line 250, correct " ≥1000 copies/ml".

7. PLOS authors have the option to publish the peer review history of their article (what does this mean?). If published, this will include your full peer review and any attached files.

Reviewer #1: No

Reviewer #2: **Yes: **Paul Spearman

---

## [Editor Report · Acceptance letter]

14 Feb 2023

PONE-D-22-26318R1 

HIV Virologic Response, Patterns of Drug Resistance Mutations and Correlates Among Adolescents and Young Adults: A Cross-Sectional Study in Tanzania 

Dear Dr. Rugemalila:

I'm pleased to inform you that your manuscript has been deemed suitable for publication in PLOS ONE. Congratulations! Your manuscript is now with our production department. 

Kind regards, 

on behalf of

Dr. Jason T. Blackard 

Academic Editor

PLOS ONE